# Disulfiram/Copper Suppresses Cancer Stem Cell Activity in Differentiated Thyroid Cancer Cells by Inhibiting BMI1 Expression

**DOI:** 10.3390/ijms232113276

**Published:** 2022-10-31

**Authors:** Yung-Lun Ni, Peng-Ju Chien, Hung-Chia Hsieh, Huan-Ting Shen, Hsueh-Te Lee, Shih-Ming Chen, Wen-Wei Chang

**Affiliations:** 1Department of Pulmonary Medicine, Taichung Tzu Chi Hospital, Buddhist Tzu Chi Medical Foundation, Taichung City 427213, Taiwan; 2Department of Biomedical Sciences, Chung Shan Medical University, Taichung City 402306, Taiwan; 3Institute of Anatomy & Cell Biology, National Yang Ming Chiao Tung University, Taipei City 112304, Taiwan; 4Bachelor Program in Health Care and Social Work for Indigenous Students, Providence University, Taichung City 433303, Taiwan; 5Department of Medical Research, Chung Shan Medical University Hospital, Taichung City 402306, Taiwan

**Keywords:** thyroid cancer, disulfiram, cancer stem cells, c-Myc, E2F1

## Abstract

Differentiated thyroid carcinomas (DTCs), which have papillary and follicular types, are common endocrine malignancies worldwide. Cancer stem cells (CSCs) are a particular type of cancer cells within bulk tumors involved in cancer initiation, drug resistance, and metastasis. Cells with high intracellular aldehyde hydrogenase (ALDH) activity are a population of CSCs in DTCs. Disulfiram (DSF), an ALDH inhibitor used for the treatment of alcoholism, reportedly targets CSCs in various cancers when combined with copper. This study reported for the first time that DSF/copper can inhibit the proliferation of papillary and follicular DTC lines. DSF/copper suppressed thyrosphere formation, indicating the inhibition of CSC activity. Molecular mechanisms of DSF/copper involved downregulating the expression of B lymphoma Mo-MLV insertion region 1 homolog (BMI1) and cell cycle-related proteins, including cyclin B2, cyclin-dependent kinase (CDK) 2, and CDK4, in a dose-dependent manner. BMI1 overexpression diminished the inhibitory effect of DSF/copper in the thyrosphere formation of DTC cells. BMI1 knockdown by RNA interference in DTC cells also suppressed the self-renewal capability. DSF/copper could inhibit the nuclear localization and transcriptional activity of c-Myc and the binding of E2F1 to the BMI1 promoter. Overexpression of c-Myc or E2F1 further abolished the inhibitory effect of DSF/copper on BMI1 expression, suggesting that the suppression of c-Myc and E2F1 by DSF/copper was involved in the downregulation of BMI1 expression. In conclusion, DSF/copper targets CSCs in DTCs by inhibiting c-Myc- or E2F1-mediated BMI1 expression. Therefore, DSF is a potential therapeutic agent for future therapy in DTCs.

## 1. Introduction

Approximately 96% of all malignancies of endocrine glands in the United States belong to thyroid cancer, and its increasing incidence rate is especially noted in Saudi females [1]. Thyroid cancer can be divided into papillary, follicular, medullary, and anaplastic types [2]. Differentiated thyroid carcinomas (DTCs) have two types, papillary thyroid carcinoma (PTC) and follicular thyroid carcinoma (FTC), with incidence rates of 80–85% and 10–15%, respectively [3].

Although PTC accounts for the majority of the incidence of thyroid cancers and is usually benign, surveillance, epidemiology, and end-results data show a 3% increase in the overall annual incidence of thyroid cancer and an increase in the mortality rate for advanced-stage PTC [4]. Advanced-stage PTC results in a one-third increase in the mortality of patients with thyroid cancer. Although both are DTCs, FTC and PTC have different prognostic factors. FTC presents more frequent distant metastasis compared with the high incidence of regional lymph node and extrathyroid metastasis in PTC at the time of diagnosis. Immobilization arising from early distant metastasis to bones caused by FTC may account for its poorer survival rate compared to PTC [5,6]. Current treatments for thyroid cancer include thyroidectomy, radioactive iodine treatment, and chemotherapy [7]. Radioactive iodine treatment is commonly used in all types of thyroid cancer but is associated with acute side effects, including ageusia, pain, and salivary gland swelling, as well as long-term side effects, such as mouth pain, dental caries, and second primary tumors [8]. Hence, developing new therapeutics without severe side effects for thyroid cancer is still required.

Growing pieces of evidence for the pathogenic role of cancer stem cells (CSCs) in thyroid cancers have been explored in the initiation phase and metastasis. Several markers, including CD44 [9], CD133 [10], aldehyde dehydrogenase (ALDH) activity, and CD326 [11], have been used to identify CSCs in thyroid cancers. In addition, the thyrospheres derived from thyroid cancer cells display increased protein expression of stemness genes, such as OCT4 [12] and SOX2 [13]. Although CSCs are a small population within cancer cells, they participate in treatment resistance, metastasis, and immune evasion. Targeting CSCs has been suggested as a promising strategy to achieve long-term benefits for patients with cancer, and many clinical trials are ongoing [14]. Disulfiram (DSF), a drug for treating alcohol abstinence, has been used clinically for a long time without major side effects [15]. The accumulation of acetaldehyde caused by DSF intake, which irreversibly inhibits all cytosolic and mitochondrial ALDH isoforms, results in the unpleasant effects of alcohol consumption. This effect contributes to the wide clinical use of DSF for alcohol abstinence. High ALDH activity could maintain sufficiently low intracellular levels of reactive oxygen species, thereby preventing CSC apoptosis due to excessive intracellular oxidative stress, enhancing tumorigenicity, and increasing migratory capacity [16]. Moreover, the anti-cancer effect of DSF is conferred by copper [17]. In anaplastic thyroid cancer cells, DSF suppresses thyrosphere formation without inhibiting ALDH1A3 activity [18]. However, whether or not DSF displays anti-CSC activity in DTCs is unknown, and its underlying molecular mechanisms in CSC-targeting activity need to be investigated. In the present study, we examined the anti-CSC potential of DSF/copper treatment in two DTC cell lines, namely, K1 cells (PTC cell line) and WRO cells (FTC cell line), and we investigated the possible underlying molecular mechanisms.

## 2. Results

### 2.1. DSF/Copper Suppresses the Cell Proliferation of DTC Cell Lines

DSF functions as an ALDH inhibitor, and ALDH activity has been used as a marker to identify thyroid CSCs [11]. In addition, the DSF/copper complex displays anti-cancer activity with low toxicity to normal cells [19]. In the present study, we examined the cytotoxicity of DSF/copper in K1 and WRO cells. Using MTT as an indicator of cell proliferation, we found that DSF/copper suppressed the proliferation of the K1 (Figure 1A) and WRO cells (Figure 1B) in a dose-dependent manner, and the half-maximal inhibitory concentrations (IC50) of DSF for the K1 and WRO cells were 18.5 ± 7.7 and 15.3 ± 3.5 μM, respectively. Targeting CSCs is considered key for the long-term control of cancer progression [14,20]. We next tested if DSF/copper inhibits the CSC activity of the DTC cell lines through thyrosphere cultivation and discovered that the suppressive effect of DSF on the thyrosphere formation of the K1 (Figure 1C) and WRO (Figure 1D) cells could be observed at a concentration of 200 nM, which was far less than the IC_50_ under adherent culture conditions. In addition, the expression of CD44, one of the markers for thyroid CSCs [21], in the K1 or WRO thyrosphere cells was obviously suppressed by DSF/copper at 200 nM (Figure 1E). These data suggest that DSF/copper could function as a CSC-targeting drug. We next examined cell proliferation using the NTP-transporter to deliver Atto-488-conjugated dUTP to the two DTC cell lines. The results showed that DSF/copper reduced Atto 488–dUTP incorporation in the K1 (Figure 2A) and WRO cells (Figure 2B) in a dose-dependent manner. We further examined the changes in cell cycle-related proteins in the DSF/copper-treated DTC cells. Results showed that DSF/copper inhibited the expression levels of cyclin B2 and cyclin-dependent kinase 6 in the K1 and WRO cells (Figure 2C). In addition, the decreased cell population in the S phase and the increased percentage of cells in the G2/M phase were observed in both K1 and WRO cells by propidium iodide (PI) staining and flow cytometric analysis (Figure 2D). These data suggest that DSF/copper reduces the proliferation of DTC cells by interrupting cell cycle progression and displays CSC-targeting activity in DTC cells.

### 2.2. BMI1 Downregulation Is Responsible for the Anti-CSC Activity of DSF/Copper in DTC Cell Lines

We extracted the total RNA from the DSF/copper-treated K1 and WRO cells and determined the mRNA expression of several cancer stemness genes, including B Lymphoma Mo-MLV Insertion Region 1 Homolog (BMI1), Homeobox Protein NANOG (NANOG), Octamer-Binding Protein 4 (OCT4), and Sex Determining Region Y-Box 2 (SOX2), by using quantitative RT-PCR to understand the molecular mechanisms through which DSF/copper suppresses thyroid CSCs. DSF/copper downregulated the mRNA expression of BMI1 (Figure 3A) and OCT4 (Figure 3B) in K1 and WRO cells in a dose-dependent manner, but the levels of SOX2 or NANOG mRNA did not show any change (Appendix A). The upregulation of BMI1 mRNA level in PTC tissues has been reported recently [22] and initiated our interest to further investigate the role of BMI1 in the anti-CSC activity of DSF/copper. Western blot analysis was used to confirm that DSF/copper inhibited the BMI1 protein expression in the K1 and WRO cells (Figure 3C). We next overexpressed BMI1 in the K1 cells through the lentiviral delivery of BMI1 cDNA, which was confirmed by Western blot (Figure 3D). The BMI1-overexpressed K1 cells were used for thyrosphere cultivation. The lentiviral vector contained a tRFP protein; therefore, we could trace the BMI1-overexpressed cells with red fluorescence. Results showed that BMI1 overexpression in the K1 cells increased the number of thyrospheres and abolished the anti-CSC effect of DSF/copper (Figure 3E). These data suggest that the anti-CSC activity of DSF/copper is mediated by BMI1 downregulation in DTC cells. We next used RNA interference to investigate the role of BMI1 in the maintenance of thyroid CSCs. The knockdown efficiencies of two BMI1-specific shRNAs or the combination of these two shRNAs were first confirmed by qRT-PCR (Figure 4A) and Western blot (Figure 4B) after transduction into the K1 cells. The primary thyrosphere formation of the K1 cells, which represents CSC activity, was observed after BMI1 knockdown in the shRNA clones or the combination of the two shRNAs (Figure 4C). We also examined the secondary thyrosphere formation of the K1 cells, which represents the self-renewal capability. The results revealed that the number of secondary thyrospheres decreased in the K1 cells transfected with sh-BMI1#1 or the two shBMI1 lentivruses (Figure 4C). These data demonstrate that DSF/copper displays anti-cancer activity in DTC cells by downregulating BMI1.

### 2.3. DSF/Copper Inhibits BMI1 Expression through c-Myc and E2F1 Downregulation

c-Myc regulates BMI1 expression in nasopharyngeal carcinoma [23]. In addition, the nuclear translocation of c-Myc in breast CSCs could maintain BMI1 expression [24]. In the present study, Western blot results showed that DSF/copper downregulated the total protein level of c-Myc in the K1 and WRO cells (Figure 5A). We further separately extracted the cytoplasmic and nuclear proteins to detect the protein level of c-Myc using Western blot. Results showed that DSF/copper inhibited the c-Myc level in the nuclear fraction of K1 and WRO cells (Figure 5B). Using a luciferase-based reporter assay involving c-Myc DNA binding elements inserted before the firefly luciferase gene, we demonstrated that DSF/copper suppressed the transcriptional activity of c-Myc in the K1 and WRO cells (Figure 5C). This result suggests that DSF/copper can potentially reduce c-Myc-induced gene expression. In addition to c-Myc, E2F1 also activates BMI1 expression in neuroblastoma [25] and gastric cancer cells [26]. Western blot analysis showed that DSF/copper downregulated E2F1 protein expression in the K1 and WRO cells (Figure 5D). Chromatin immunoprecipitation with the anti-E2F1 antibody in K1 cells revealed that DSF/copper reduced the binding of E2F1 on the BMI1 promoter (Figure 5E). Regarding the treatments of HLM-006474, an E2F1 inhibitor, or 10058-F4, a c-Myc inhibitor, suppressed BMI1 protein expression in the K1 and WRO cells (Figure 5F). The overexpression of c-Myc or E2F1 in WRO cells abolished the inhibitory effect of DSF/copper on BMI1 protein expression (Figure 5G). These data strongly support that DSF/copper downregulates BMI1 expression in DTC cells by downregulating c-Myc or E2F1 expression. We further analyzed the thyroid carcinoma (THCA) dataset of the Cancer Genome Atlas (TCGA) database, which is a collection of 512 PTC samples, including 272 BRAF-like and 118 RAS-like PTCs. The results revealed that the mRNA expression of c-Myc (Figure 6A) and E2F1 (Figure 6B) displayed a remarkably positive correlation with BMI1. Using the median BMI1 mRNA expression level as a cutoff value to perform gene set enrichment analysis, we found that BMI1 expression in the PTC cells was positively correlated with the gene sets of the E2F target (Figure 6C), the G2M checkpoint (Figure 6D), and the mitotic spindle (Figure 6E) but was negatively correlated with the gene set of oxidative phosphorylation (Figure 6F). These data suggest that the inhibitory activity of DSF/copper in BMI1 expression is mediated by the reduction in c-Myc and E2F1 expression, and the BMI1 expression in PTC may be highly correlated with its malignant phenotype.

## 3. Discussion

The present study has some limitations. It only included two DTC cell lines and lacked a xenograft mouse model of DTC to examine the anti-cancer efficacy of DSF/copper in vivo. Furthermore, the use of BMI1 non-expressing cell lines may be required to confirm the association between BMI1 and the anti-cancer effect of DSF/copper. DTCs account for 90% of all thyroid cancers. Although they can be well controlled by radioactive iodine, 6–20% of patients with DTC still experience recurrence, and the absorption of radioactive iodine via the recurrence of cancer lesions becomes poor [28]. BMI1 has been demonstrated to participate in cancer radioresistance. BMI1 participates in cancer radioresistance. For example, BMI1 knockdown in glioblastoma multiforme cells impairs the activation of ataxia telangiectasia-mutated kinase and sensitizes the cells to radiation [29]. The radioresistant function of BMI1 was also observed in colorectal cancer cells, and the overexpression of Kruppel-like factor 4 partially confers the radiosensitized phenotype in BMI1-knockdown colorectal cancer cells [30]. In the present study, DSF/copper inhibited BMI1 expression (Figure 3). The involvement of BMI1 in the resistance of radioactive iodine and the sensitization potential of DSF/copper in DTCs with radioactive iodine resistance is worthy of further investigation.

In addition to radioactive iodine, several targeted anti-cancer drugs that are based on multitargeted kinase inhibitors, such as lenvatinib or sorafenib, have been approved for radioiodine-refractory DTCs [31]. However, drug resistance against kinase inhibitors is frequently observed among patients with cancer. The activation of epidermal growth factor receptor (EGFR) could limit the efficacy of lenvatinib in hepatocellular carcinoma (HCC) [32]. Ngo et al. reported that the expression of insulin-like growth factor-1 (IGF1R) is upregulated in sorafenib-resistant HCC cells and that IGF1R activation causes the nuclear translocation of Yes-associated protein followed by the induction of sorafenib resistance [33]. Using the TCGA database, we further discovered that the gene set of EGFR_UP.V1_UP [34] was enriched among patients with THCA and high BMI1 expression (Appendix A), and a remarkably positive correlation was found between BMI1 and EGFR in the THCA dataset (Appendix A). We also found the enrichment of the EGFR_UP.V1_DN gene set [34] in patients with THCA and high BMI1 expression (Appendix A) and a considerably positive correlation between BMI1 and IGF1R (Appendix A). The BMI1 inhibition function suggests that DSF/copper could be a combination drug for preventing the drug resistance of multitargeted kinase inhibitors in thyroid cancers.

c-Myc is an important oncogene that contributes to carcinogenesis, and approximately 28% of human cancers have amplified MYC paralogs [35], which could be an attractive target for drug development. However, the development of direct inhibitors for c-Myc is challenging because of the lack of a specific active site for the binding of small molecules, and the nuclear localization characteristics complicate the delivery of neutralization antibodies [36]. E2F1 is overexpressed in many cancers, including non-small cell lung cancer [37], gastric cancer [38], and papillary and anaplastic thyroid cancers [39]. In addition to cell cycle regulation, E2F1 has been reported to participate in the maintenance of CSCs by directly transcriptionally activating cancer stemness factors, including NANOG [40] and Kruppel-Like Factor 4 [41]. The developed inhibitors for E2F1 include the disruption of E2F1/DP1 binding by peptides or oligonucleotide decoys, but the main issue is the poor uptake by tumor cells [42]. The present data demonstrated that DSF/copper can inhibit the protein expression of c-Myc and E2F1, the nuclear translocation of c-Myc, and the binding of E2F1 on the BMI1 promoter (Figure 5). Therefore, DSF/copper could be a potential drug for cancers with c-Myc or E2F1 overexpression. O’Donnell et al. previously demonstrated that c-Myc can modulate E2F1 expression by regulating the expression of microRNAs that target E2F1, including miR-17-5p and miR-20a [43]. In the present study, we did not confirm a direct association between c-Myc and E2F1 in the DTC cell lines, and this topic warrants further examination in the future.

Skrott et al. performed a nationwide epidemiological study to investigate cancer incidence among continuing DSF users in Denmark and found reduced overall cancer mortality (hazard ratio = 0.66, 95% confidence interval = 0.58–0.76, *p* = 0) [44]. After consumption, DSF is rapidly converted into methylated diethyldithiocarbamate ditiocarb and could be complexed with copper, which could also be detected in the tumor tissues of continuing DSF users [44]. DSF is an approved drug for treating alcoholism and is well tolerated among patients without major side effects [45]. Molecular analysis in the present study showed that DSF/copper induces its anti-DTC effect by inhibiting BMI1 expression and suppressing CSC activity. CSCs are important in cancer recurrence and drug resistance; hence, DSF could be a promising repurposed drug that could be used in combination with current managements, including radioactive iodine or targeted therapeutics, for DTC therapy.

## 4. Materials and Methods

### 4.1. Cell Culture

K1 cell line belongs to PTC, was purchased from the European Collection of Cell Cultures (ECACC, UK Health Security Agency, Salisbury, UK), and was maintained in DMEM:Ham’s F12:MCDB 105 (2:1:1) medium containing 10% fetal bovine serum (FBS, purchased from HyClone™ Laboratories, Cytiva, Marlborough, MA, USA). The WRO cell line belongs to FTC, was also purchased from ECACC, and was maintained in DMEM medium containing 10% FBS.

### 4.2. Chemicals and Reagents

DSF was purchased from Sigma-Aldrich (St. Louis, MI, USA) and dissolved in dimethyl sulfoxide (DMSO) at a concentration of 50 mM stored at −20 °C. Blasticidin S or puromycin was purchased from TOKU-E (Bellingham, WA, USA) and dissolved in PBS at 20 mg/mL and stored at −20 °C. The aminoallyl-XX-dUTP-Atto-488 triethylammonium salt solution (Atto-488 conjugated dUTP) was purchased from Sigma-Aldrich. HLM 006474, an E2F1 inhibitor, and 10058-F4, a c-Myc inhibitor, were purchased from Tocris Biosciences (Bristol, UK).

### 4.3. WST-1 Based Cell Proliferation Assay

K1 or WRO cells were seeded into wells of 96-well-plates as 1 × 10^4^ cells/well and treated with different concentrations of DSF in the presence of 1 μM CuCl_2_. The concentration referred to the report of Wang et al. [46], after culturing for 72 h. The cell proliferation was monitored by adding the WST-1 reagent (BioVision, Inc., Waltham, MA, USA) at 10 μL/well and incubating it at 37 °C for 1 h followed by determination of the absorbance at 440nm wavelength. The IC_50_ values were calculated by GraFit (version 5.0.6, Erithacus Software, West Sussex, UK).

### 4.4. Thyrosphere Cultivation

K1 or WRO cells were seeded into wells of an ultralow attachment surface 6-well-plate (Greiner Bio-One GmbH, Kremsmünster, Austria) at a cell number of 2 × 10^4^ cells/well with DMEM/F12 media containing 0.4% bovine serum albumin (Sigma-Aldrich), 1X B27 supplement (Gibco^TM^, Waltham, MA, USA), 10 ng/mL EGF (PeproTech, Inc., Rocky Hill, NJ, USA), 10 ng/mL bFGF (PeproTech), 5 μg/mL Insulin (Sigma-Aldrich), 1 μg/mL Hydrocotisone (Sigma-Aldrich), and 4 μg/mL heparin (Sigma-Alrich) at 37 °C for 7 days to form primary thyrospheres. Secondary thyrosphere cultivation was performed by collecting the primary thyrospheres and dissociating them into single-cell suspension by HyQTase solution (Hyclone Laboratories, Inc., Logan, UT, USA).

### 4.5. Determination of Cell Proliferation by the Incorporation of Atto-488 Conjugated dUTP

The K1 or WRO cells were seeded into wells of 24-well plates at 1 × 10^5^ cells/well and cultured at 37 °C overnight followed by washing with Tricine buffer (Tricine 5 mM, glucose 11 mM, NaCl 125 mM, CaCl_2_ 1.8 mM, MgSO_4_ 0.8 mM, KCl 5.4 mM). Then, 10 μM of Atto-488 conjugated dUTP was mixed with 10 μM of BioTracker^TM^ NTP-transporter at room temperature for 10 min and added into wells for 10 min followed by washing with 0.4 mL PBS once. After adding 0.5 mL of the complete culture medium, the green fluorescence signals were captured by inverted fluorescent microscopy (AE31 ELITE, Motic Asia, Kowloon, Hong Kong). The percentage of Atto-488 dUTP+ cells was counted from 4 random fields using the Cell Counter function of Image J software (version 1.8.0_172, National Institutes of Health, Bethesda, MA, USA).

### 4.6. Flow Cytometrical Analyses of Cell Cycle Distributions or CD44 Expression

For cell-cycle analysis, cells were seeded in wells of 6-well plates at 1 × 10^5^ cells/well and treated with 100 nM DSF in the presence of 1 μM CuCl2 at 37 °C/5% in a CO_2_ incubator for 24 h. After harvesting via trypsin/EDTA, cells were then fixed with 70% EtOH/PBS at 4 °C overnight. After washing with PBS once, they were stained with 10 μg/mL PI/PBS solution in the presence of 10 μg/mL RNaseA (Cat. No. 550825, BD Biosciences, Franklin Lakes, NJ, USA) at room temperature for 30 min. For CD44 detection, thyrospheres derived from K1 or WRO cells were first dissociated into single-cell suspensions by enzyme-free cell dissociation buffer (Cat. No. 13151014, Gibco^TM^, Thermo Fisher Scientific) and stained with PE-conjugated mouse IgG anti-human CD44 antibody (Cat. No. 550989, BD Biosciences) on ice for 30 min. The fluorescence signals of PI or PE were captured with the FACSCanto-II flow cytometer (BD Biosciences). Finally, the cell cycle distributions or CD44 expression were analyzed with FlowJo software (version 10.8.0, FlowJo, LLC., Ashland, OR, USA).

### 4.7. Quantitative RT-PCR

Total cellular RNA was extracted by the Quick-RNA^TM^ MiniPrep Kit (Cat. No. R1054, Zymo Research Corporation, Irvine, CA, USA) and was reverse transcribed into complementary DNA (cDNA) by the RNA RevertAid First Strand cDNA Synthesis Kit (Thermo Fisher Scientific, Waltham, MA, USA). Then, 10 ng cDNA were used for gene expression analysis using the SYBR Green Master Mix (Bio-Rad Laboratories, Hercules, CA, USA) and an Eco 48 real-time PCR system (PCR max, Staffordshire, UK) with specific qPCR primer pairs whose sequences are listed in Appendix A.

### 4.8. Manipulation of BMI1 Expression Levels in K1 Cells

For the knockdown of BMI1 expression, lentiviruses carrying BMI1-specific shRNA sequences (TRCN0000020156, which was indicated as sh-BMI1#1, and TRCN0000229416, which was indicated as sh-BMI1#2, all obtained from the National RNAi Core Facility at Academia Sinica, Taipei, Taiwan) were produced in 293T cells and were transduced into K1 cells followed by the selection of 2 μg/mL puromycin according to the protocol described in our previous report [47]. The lentiviral vector carrying BMI1 cDNA has been established in our laboratory as in the previous report [47]. The successfully transduced K1 cells were selected by 20 μg/mL blasticidin S and were further isolated according to the tRFP fluorescence signals by the FACS Aria-II cell sorter (BD Biosciences).

### 4.9. Western Blot Analysis

Cells were lysed by RIPA buffer (Thermo Fisher Scientific Inc.) and the total protein concentrations were quantified by the Pierce™ BCA Protein Assay Kit (Thermo Fisher Scientific Inc., Cat. No. 23227). The cytoplasmic or nuclear fractions of cells were extracted by a Subcellular Protein Fractionation Kit (Thermo Fisher Scientific Inc., Cat. No. 78840). Furthermore, 20 μg of extracted proteins were separated by SDS-PAGE and were transferred onto a Polyvinylidene fluoride (PVDF) membrane (Pall Corporation, Washington, NY, USA). After blocking with 10% skim milk in TBS-T buffer (20 mM Tris-HCl, 150 mM NaCl, 0.05% Tween-20) at room temperature for 1 h, the PVDF membrane was then incubated with the primary antibody at 4 °C overnight followed by incubation with horseradish peroxidase-conjugated secondary antibody. The antibodies used in this study are listed in Appendix A. The signals were then developed via incubation with the Pierce™ ECL Western Blotting Substrate (Thermo Fisher Scientific Inc.) and captured with the LAS-4000 Luminescence Image System (GE Healthcare, Chicago, IL, USA). The band intensities were quantitated by Image J software.

### 4.10. Chromatin Immunoprecipitation

The preparation of Chromatin DNA from K1 cells was conducted according to the protocol described in our previous report [48]. First, 75 μg of chromatin DNA was used for incubating with 1 μg of the anti-E2F1 antibody (BD Pharmingen™) in dilution buffer (167 mM NaCl, 16.7 mM Tris (pH8.1), 1.2 mM EDTA, 1.1% Triton X-100, 0.01% SDS) at 4 °C overnight. After the washing steps described in a previous report [21], the RNA and proteins were digested by RNaseA and proteinase K followed by the extraction of DNA using the QIAquick PCR purification kit (Cat. No. 28104, QIAGEN, Hilden, Germany). The presence of the BMI1 promoter DNA sequence was detected by quantitative PCR with the specific primer set listed in Appendix A. Data are presented as the relative percentage of input chromatin.

### 4.11. c-Myc Reporter Assay

A luciferase-based c-Myc reporter vector, pMyc-Luc, was purchased from Signosis Inc. (Cat. No. LR-2018, Signosis Inc., Santa Clara, CA, USA) and was mixed with the pRL vector, carrying the Renilla luciferase gene to adjust the transfection efficiency, at a ratio of 1 μg: 0.1 μg followed by complexing with Transit X2^TM^ transfection reagent at room temperature for 15 min. After seeding cells in wells of 12-well-plates as 1 × 105 cells/well and allowing attachment overnight, the DNA/transfection reagent complex was inserted into cells for 24 h followed by changing to fresh culture media and adding DSF/copper for a further 48 h. Total cell lysates were harvested with passive lysis buffer (Promega Corporation, Madison, WI, USA), and the luciferase activities of firefly and renilla were measured by a Dual-Luciferase assay system (Promega) with a GloMax^®^ 20/20 Luminometer (Promega).

### 4.12. Overexpression of c-Myc or E2F1

WRO cells were seeded into 3.5 cm dishes as 2 × 105 cells/dish and transfected with the pCMV3 empty vector or plasmids carrying cDNA of c-Myc (Cat. No. HG11346-NY, Sino Biological, Beijing, China) or E2F1 (Cat. No. HG17394-UT, Sino Biological) using Transit X2 transfection reagent at 37 °C for 24 h. After treatment with DSF/copper for 48 h, cells were harvested by trypsin/EDTA and lysed by the NETN buffer.

### 4.13. Statistical Analysis

The values of qRT-PCR, cell proliferation, and thyrospheres were expressed as the mean ± standard deviation (SD). The significance between the two groups was calculated by Student’s t-test and the difference between multiple groups was calculated by the One-Way ANOVA with Tukey–Kramer’s post hoc test using Prism (version 5.0, GraphPad Software, San Diego, CA, USA).

## 5. Conclusions

DSF/copper displays an anti-CSC activity in DTC cell lines by suppressing BMI1 expression. DSF/copper targets thyroid CSCs by downregulating nuclear c-Myc and inhibiting the binding of E2F1 to the BMI1 promoter. Toxicologic and pharmacologic data suggest that DSF/copper is a potential therapeutic agent for DTCs.

## Figures and Tables

**Figure 1 ijms-23-13276-f001:**
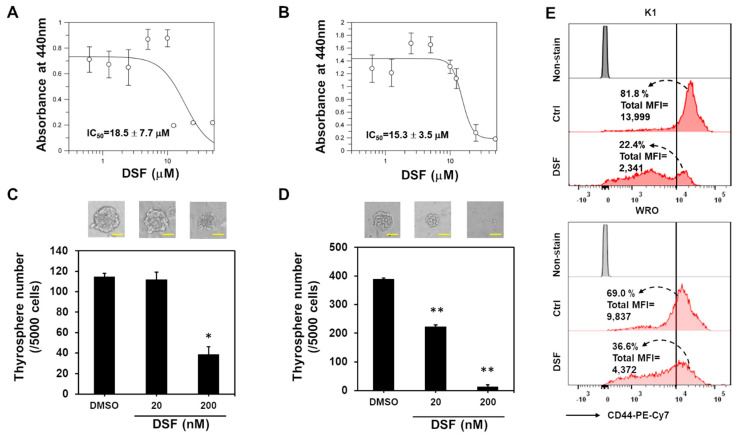
DSF/copper inhibits cell proliferation and thyrosphere formation of DTC cells. (**A**,**B**) K1 (**A**) or WRO (**B**) cells were seeded into 96-well-plates and treated with the indicated concentrations of DSF in presence of 1 μM CuCl_2_ for 72 h. The cell proliferation was determined by WST-1 reagent and the absorbance was read at 440 nm wavelength. The curves and IC_50_ values were drawn and calculated with GraFit software. (**C**,**D**) K1 (**C**) or WRO (**D**) cells were seeded into ultralow attachment culture dishes to form primary thyrospheres for 7 days under the treatment of indicated concentrations of DSF in presence of 1 μM CuCl_2_. The formed thyrospheres were pictured and counted on Day 7. *, *p* < 0.05; **, *p* < 0.01. Scale bars in (**C**,**D**) represented as 50 nm. (**E**) The thyrospheres from K1 or WRO cells were dissociated into single-cell suspensions by enzyme-free cell dissociation buffer and treated with 200 nM of DSF in presence of 1 μM CuCl_2_ for 24 h. The expression of CD44 was determined by flow cytometry. Data were analyzed by FlowJo software. The dotted arrows indicate the percentage of CD44 high expression cells that with the fluorescence intensity greater than 10^4^.

**Figure 2 ijms-23-13276-f002:**
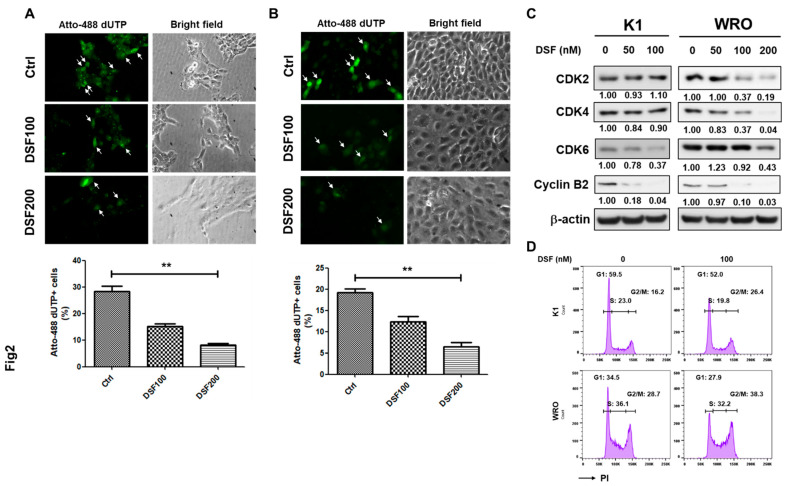
DSF/copper inhibits cell cycle progression in DTC cells. (**A**,**B**) K1 (**A**) or WRO (**B**) cells were seeded into 12-well-plates and treated with indicated concentrations of DSF in presence of 1 μM CuCl_2_ for 48 h followed by adding Atto-488 dUTP mixing with BioTrackerTM NTP-transporter for 10 min. The uptake of Atto-488 dUTP was determined by green fluorescence capture (as indicated by arrows) and counted with ImageJ software. **, *p* < 0.01. Scale bars: 20 μm. (**C**) K1 or WRO cells were treated with the indicated concentrations of DSF for 48 h and the protein expressions of CDK2, CDK4, CDK4, and cyclin B2 were determined by Western blot. Inserted numbers indicated the relative expression levels in comparison with DSF non-treated cells. (**D**) K1 or WRO cells were treated with 100 nM DSF in presence of 1 μM CuCl_2_ for 24 h. Cells were then harvested and fixed with 70% EtOH/PBS, stained with PI in presence of RNaseA, and the fluorescence signals were collected by flow cytometry. The cell cycle distributions of each cell line were quantified by FlowJo software.

**Figure 3 ijms-23-13276-f003:**
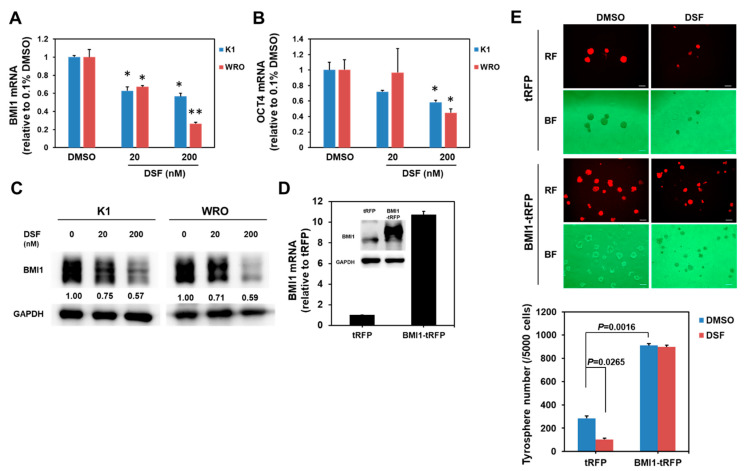
DSF/copper downregulates BMI1 expression. (**A**,**B**) Total RNA of DSF/copper-treated K1 or WRO cells was extracted and reverse-transcribed into cDNA. The mRNA expressions of BMI1 (**A**) or OCT4 (**B**) were determined by SYBR Green-based real-time PCR. *, *p* < 0.05; **, *p* < 0.0.1. (**C**) The BMI1 protein expression of K1 or WRO cells under DSF/copper treatments was determined by Western blot analysis. Inserted numbers indicate the relative expression level in comparison with DSF non-treated cells. (**D**) K1 cells were transduced with tRFP- or BMI1-carrying lentiviruses and selected with 20 μg/mL blasticidin for 96 h. The mRNA or protein expressions of BMI1 were determined by real-time RT-PCR or Western blot, respectively. (**E**) The CSC activity of tRFP or BMI1 overexpressed K1 cells under DSF/copper treatment at 200 nM was determined by thyrosphere formation capability. The formed thyrospheres were pictured and counted on Day 7. RF, red fluorescence; BF, bright field. The white lines in each picture indicated a scale bar of 100 μm.

**Figure 4 ijms-23-13276-f004:**
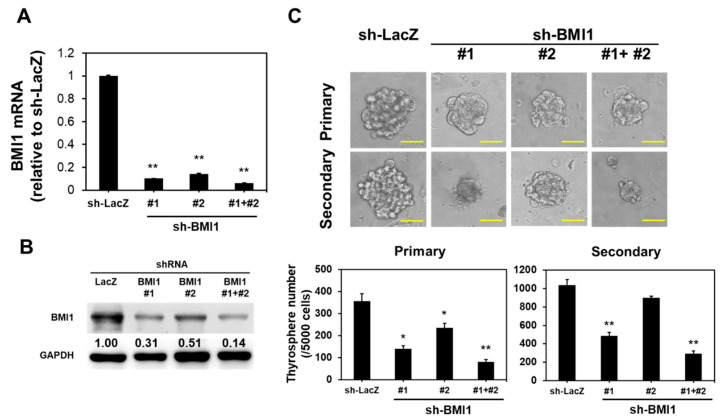
Knockdown of BMI1 suppresses CSC activity in K1 cells. K1 cells were transduced with lentiviruses carrying BMI1-specific shRNAs (shBMI1#1, shBMI1#2, or the combination #1plus #2 as a ratio of 1:1) followed by selection with 2 μg/mL puromycin for 72 h. (**A**,**B**) The mRNA (**A**) or protein (**B**) expressions of BMI1 were determined by real-time RT-PCR or Western blot, respectively. (**C**) The shRNA-transduced K1 cells were used for primary thyrosphere cultivation and counted the sphere numbers on Day 7. After counting, the formed primary thyrospheres were collected with 100 μm cell strainers and dissociated into single-cell suspension after treatment with HyQTase followed by secondary thyrosphere cultivation and counting at Day 7 post seeding. Scale bars: 50 μm. *, *p* < 0.05; **, *p* < 0.01.

**Figure 5 ijms-23-13276-f005:**
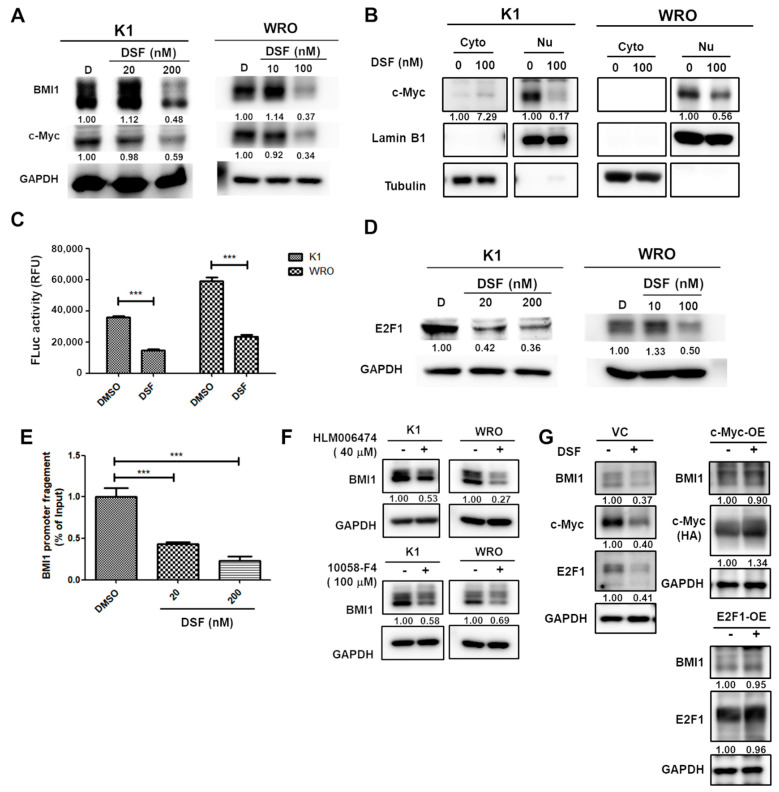
DSF/copper reduces c-Myc and E2F1 activation in DTC cells. (**A**,**B**) Total cellular proteins (**A**) or the cytoplasmic (Cyto)/nuclear (Nu) fractions (**B**) of K1 or WRO cells after DSF/copper treatment were extracted and the c-Myc protein was determined by Western blot. Lamin B1 or tubulin in (**B**) was used as the marker for nuclear or cytoplasmic proteins, respectively. (**C**) The E2F1 protein expression in total cellular proteins from DSF/copper treated K1 or WRO cells was determined by Western blot. (**D**) pMyc-Luc and pRL vectors were mixed as a ratio of 100:1 and were transfected into K1 or WRO cells for 24 h followed by treatment of 100 nM DSF in presence of 1 μM CuCl_2_ for further 48 h. After harvesting the total cell lysates with passive lysis buffer, firefly luciferase or activities were then measured, and data were normalized with Renilla luciferase activity of each sample. ***, *p* < 0.001. (**E**) Chromatin DNA of DSF/copper treated K1 cells was extracted and performed immunoprecipitation with anti-E2F1 antibody. The BMI1 promoter DNA fragments within precipitated chromatins were then quantitated by real-time PCR method. The data were presented as relative percentage of input DNA. ***, *p* < 0.001. (**F**) K1 or WRO cells were treated with HLM006474 or 10058-F4 at 40 μM or 100 μM, respectively, for 48 h and the protein expression of BMI1 was determined by Western blot. The inserted numbers indicated relative expression levels compared to non-treated samples. (**G**) WRO cells were transfected with pCMV3 (VC) or vectors carrying cDNA of c-Myc (c-Myc-OE) or E2F1 (E2F1-OE) for 24 h followed by treatment with 100 nM DSF in presence of 1 μM CuCl_2_ for further 48 h. The protein expressions of BMI1, c-Myc, or E2F1 were determined by Western blot. For c-Myc overexpression, exogenous c-Myc was tagged with hemagglutinin (HA) and was detected by anti-HA antibody. The inserted numbers indicated relative expression levels compared to non-DSF treatment samples.

**Figure 6 ijms-23-13276-f006:**
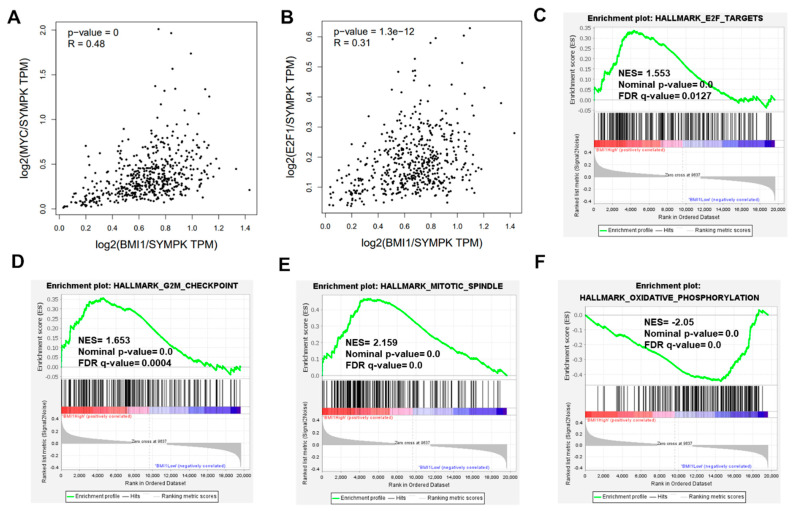
The analysis of the enriched gene sets based on BMI1 expression level in THCA dataset of TCGA database. (**A**,**B**) The correlations of BMI1 with c-Myc or E2F1 in THCA dataset of TCGA database were obtained from GEPIA_2 website (http://gepia2.cancer-pku.cn, accessed on 20 June 2022). SYMPK was chosen as a housekeeping gene for normalization [27] (**C**–**F**) The gene set enrichments of THCA dataset were analyzed by GSEA software according to the BMI1 mRNA expression level using the median as a cut-off value.

## Data Availability

Not applicable.

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
