# Peer review of "Disulfiram/Copper Suppresses Cancer Stem Cell Activity in Differentiated Thyroid Cancer Cells by Inhibiting BMI1 Expression"

_ijms, 2022, doi:10.3390/ijms232113276_

Round 1

Reviewer 1 Report

Ni et al showed that Disulfiram/copper suppresses cancer stem cell activity in differentiated thyroid cancer cells by inhibiting BMI1 expression. The manuscript overall is not clearly laid out and numerous grammar errors exist throughout the paper. On balance, the results presented are not convincing either. From my perspective, there are multiple major concerns that should be addressed.

1) Fig 1c-d, fig.2a-b, fig.3e, fig.4c, lack scale bar.

2) Running flow cytometry to access the cell cycle will be more accurate.

3) Fig.3 , have you checked other CSC markers? such as SOX2, Nanog, or known CSC surface markers.

4) Except  secondary sphere forming capability, more readouts for CSC activity should be evaluated.

5) At least do a simple in vitro study to show the tumor suppressive function of disulfiram/copper. 

The authors are also strongly recommended to seek professional English language editing service to improve the manuscript. 

Author Response

1) Fig 1c-d, fig.2a-b, fig.3e, fig.4c, lack scale bar.

Responses:

We appreciate the suggestions from the reviewer. There were scale bars in Fig. 4C in the original submission. The scale bars for Fig. 1c-d, Fig. 2a-b, and Fig. 3e have been added in the revision form.

2) Running flow cytometry to access the cell cycle will be more accurate.

Responses:

We appreciate the suggestions from the reviewer. The cell cycle analysis data were added in Figure 2D, which showed that DSF/copper decreased the percentage of cells in the S phase and increased the percentage of cells in the G2/M phase in both K1 and WRO cells. These data also fit the observations in the downregulation of cyclin B1, CDK4, CDK6, and CDK2, as shown in Figure 2C.

3) Fig.3 , have you checked other CSC markers? such as SOX2, Nanog, or known CSC surface markers.

Responses:

We appreciate the suggestions from the reviewer. We have already checked the mRNA expressions of SOX2 or Nanog, as shown in the Supporting data of Figure S1 and these two genes were not suppressed by DSF/copper treatment. In this revision, we further provided the downregulation of CD44 in both K1 and WRO cells using flow cytometric analysis (Fig. 1E). We hope these data could convince the reviewer that DSF/copper truly inhibits the CSC activity of DTC cells.

4) Except  secondary sphere forming capability, more readouts for CSC activity should be evaluated.

Responses:

We appreciate the suggestions from the reviewer. We have checked the level of CD44 after DSF/copper treatment and found that it could be suppressed. The data were provided as a new Figure 1E in this revision.

5) At least do a simple in vitro study to show the tumor suppressive function of disulfiram/copper. 

Responses:

We have provided the data of in vitro cytotoxicity analysis of DSF/copper in DTC cells with MTT assay (Figure 1A and 1B). More importantly, DSF/copper treatment could inhibit cancer stem cell (CSC) activity using tumorsphere assay and the downregulation of CD44, a CSC marker for thyroid cancer (ref 19 in this revision).

6) The authors are also strongly recommended to seek professional English language editing service to improve the manuscript. 

Responses:

We appreciate the suggestions from the reviewer. The revised version of the manuscript has been edited by a native speaker, and a certificate was included as supporting material.

Reviewer 2 Report

In this paper, entitled “Disulfiram/copper suppresses cancer stem cell activity in differentiated thyroid cancer cells by inhibiting BMI1 expression” by Yung-Lun Ni, it is suggested that inhibition of BMI1 expression by  Disulfiram/copper suppress cancer stem cells activity in DTCs. The study is well written, and easy to follow I like the set of experiments the authors decided to perform. This is overall a very interesting and well designed study that will contribute to our understanding of the relationship between BMI1 and cancer stem cell propriety in thyroid cancers and should be published.

Suggestion to the authors

In order to bring new knowledge to the field, the suggested treatment modality should also be tested in-vivo experimental models. Still, clinical data is missing.
If I can point an aspect that could be improved, if the treatment reduce the cancer stem cell propriety, it will be interesting to check whether it can also sensitive cancer cells to the current standard treatment.
Figure 5: From the mechanistic point of view in order to prove a direct involvement of c-Myc the author should knock-down c-Myc and show the reduced binding of E2F on BMI1 promoters.

Author Response

In this paper, entitled “Disulfiram/copper suppresses cancer stem cell activity in differentiated thyroid cancer cells by inhibiting BMI1 expression” by Yung-Lun Ni, it is suggested that inhibition of BMI1 expression by  Disulfiram/copper suppress cancer stem cells activity in DTCs. The study is well written, and easy to follow I like the set of experiments the authors decided to perform. This is overall a very interesting and well designed study that will contribute to our understanding of the relationship between BMI1 and cancer stem cell propriety in thyroid cancers and should be published.

Responses:

We appreciate the positive feedback from the reviewer, and the point-by-point responses are listed below.

Suggestion to the authors

1. In order to bring new knowledge to the field, the suggested treatment modality should also be tested in-vivo experimental models. Still, clinical data is missing.

Responses:

We appreciate the suggestions from the reviewer. Due to the limited time for revisions, we cannot perform the in vivo tumor engraftment assay within 10 days. However, we have provided the results of analyzing the Cancer Genome Atlas (TCGA) database that support the importance of c-Myc or E2F1 in the BMI1 expression level (Fig. 6). We think the data from TCGA could support our molecular examinations using DTC cell lines. In response to the comments, we add some descriptions to present the limitations of this study in the first paragraph of the Discussion section as follows: “The present study has some limitations. It included only two DTC cell lines and lacked a xenograft mouse model of DTC to examine in vivo the anti-cancer efficacy of DSF/copper. Furthermore, the use of BMI1 non-expressing cell lines may be required to confirm the association between BMI1 and anti-cancer effect of DSF/copper.” (page 13, line 295-298)

  1. If I can point an aspect that could be improved, if the treatment reduce the cancer stem cell propriety, it will be interesting to check whether it can also sensitive cancer cells to the current standard treatment.

Responses:

We appreciate the suggestions from the reviewer. The current treatments for DTCs include surgery and radioactive iodine treatment. Due to the limited time for revisions, we can not perform the radioactive iodine treatment, but we have made some discussions about the link between BMI1 expression and radiosensitivity in the first paragraph of the Discussion Section (page 13, line 299-311).

  1. Figure 5: From the mechanistic point of view in order to prove a direct involvement of c-Myc the author should knock-down c-Myc and show the reduced binding of E2F on BMI1 promoters.

Responses:

We apologize for the unclear explanations in Figure 5. Actually, we think the suppressive effect of DSF/copper on BMI1 expression may achieve through two possibilities, including the inhibition of c-Myc and the downregulation of E2F1. Our study did not confirm a direct association between c-Myc and E2F1 in DTC cell lines. However, a previous study demonstrated that c-Myc could modulate E2F1 expression by regulating the expression of microRNA that could target E2F1. In this revision, we add some discussions in the fourth paragraph of the Discussion section as follows: “O'Donnell et al. previously demonstrated that c-Myc can modulate E2F1 expression by regulating the expression of microRNAs that target E2F1, including miR-17-5p and miR-20a [41]. In the present study, we did not confirm a direct association between c-Myc and E2F1 in the DTC cell lines, and this topic warrants further examination in the future.” (Page 13, lin347-page 14, line 351).

In addition, we performed the treatment of HLM006474, an E2F1 inhibitor, or 10058-F4, a c-Myc inhibitor, followed by examining the expression of BMI1 (Fig. 5E). Both inhibitors caused the downregulation of BMI1 protein expression (Fig. 5E). We also overexpressed c-Myc or E2F1 in WRO cells followed by DSF/copper treatment and found that DSF/copper lost its suppressive effect in BMI1 expression (Fig. 5G). In addition, BMI1 could be upregulated when overexpressed c-Myc (Fig. 5G). Altogether, the downregulation of c-Myc or E2F1 participates in DSF/copper-mediated BMI1 suppression in DTC cells.

Reviewer 3 Report

In the present manuscript, Ni et al. investigated the ability of Disulfiram (DSF), in combination with copper, to target cancer stem cells (CSC) in differentiated thyroid carcinomas (DTC). They found  that DSF/copper inhibited cell proliferation of K1 and WRO cell lines and suppressed thyrosphere formation. The authors provided some evidences indicating that DSF/copper inhibited the expression of B lymphoma Mo-MLV insertion region 1 homolog (BMI1) gene and a number of cell cycle-related proteins, in a dose-dependent manner. To study the molecular mechanisms by which BMI1 down-regulation was involved in the inhibition of DSF/copper-mediated thyrosphere formation, the authors overexpressed BMI1 in K1 and WRO cells and found that up-regulated BMI1 counteracted the inhibitory effect of DSF/copper in the thyrosphere formation. On the contrary, BMI1 knockdown in the same DTC cells suppressed their self-renewal capability. Finally, the authors demonstrated that DSF/copper-mediated BMI1 inhibition was determined by the suppression of c-Myc and E2F1 transcriptional activity on the BMI1 promoter.

The manuscript is interesting, but a number of issues should be addressed by the authors to make it suitable for publication on IJMS.

1.      First of all, the authors limited their observations on the effect of DSF/copper on thyrosphere to only two DTC cell lines, K1 papillary thyroid carcinoma cell line and WRO follicular thyroid carcinoma cell line. They are strongly encouraged to extend their analyses to additional DTC cell lines such as TPC-1, NIM-1, FB-2, BCPAP (papillary thyroid carcinoma -PTC- cell lines) and FTC-133, FTC-236, FTC-238 (follicular thyroid carcinoma -FTC- cell lines). Extending their study on other cell lines could be useful to acquire information about the role of activated RET/PTC (TPC-1 and FB-2 cell lines) and BRAF (BCPAP cell line) on the effects mediated by DSF/copper on thyrospheres. In particular, since PTCs carrying BRAF V600E mutation are more aggressive than RET/PTC positive carcinomas, it would be interesting to analyze the ability of DSF/copper to interfere with tumor progression of more aggressive PTCs.

2.      In figure 3E the authors showed that BMI1 over-expression in K1 cells made them resistant to DSF/copper treatment. Even so data are solid, BMI1 over-expression in a cell line showing high levels of the endogenous protein (Fig. 3C) could be a bit stretched. The reviewer strongly suggests to over-express BMI1 in a BMI1-negative PTC or FTC cell line in order to compare the contribution of BMI1 to the resistance to DSF/copper respect to a BMI1-free cellular environment.

3.      In figure 5 c-Myc and E2F1 experiments were performed at different DSF concentrations in K1 and WRO cells: why? To make a correct and significative comparison, DSF concentrations must be the same in all experiments. In addition, in K1 cells E2F1 seems to be more sensitive than c-Myc to drug effect suggesting that the contribution of E2F1 to BMI1 transcription in K1 cells could be more relevant. To this end, the authors should perform a ChIP assay also for c-Myc in K1 cells rather than cyto/nucleus fractionation, at the same DSF concentration of Western blot depicted in panel A, and they should add 20 nM DSF in E2F1 ChIP given that Western blot showed a E2F1 decrease at 20 nM similar to 200 nM. Finally, given the possible main role of E2F1, the authors should use specific c-Myc and E2F1 inhibitors to establish the contribution of these two transcription factors to BMI1 expression in K1 cells.

4.      Which types of thyroid carcinomas were considered in figure 6? The authors do not specify if they were papillary, follicular or anaplastic carcinomas.

Author Response

In the present manuscript, Ni et al. investigated the ability of Disulfiram (DSF), in combination with copper, to target cancer stem cells (CSC) in differentiated thyroid carcinomas (DTC). They found  that DSF/copper inhibited cell proliferation of K1 and WRO cell lines and suppressed thyrosphere formation. The authors provided some evidences indicating that DSF/copper inhibited the expression of B lymphoma Mo-MLV insertion region 1 homolog (BMI1) gene and a number of cell cycle-related proteins, in a dose-dependent manner. To study the molecular mechanisms by which BMI1 down-regulation was involved in the inhibition of DSF/copper-mediated thyrosphere formation, the authors overexpressed BMI1 in K1 and WRO cells and found that up-regulated BMI1 counteracted the inhibitory effect of DSF/copper in the thyrosphere formation. On the contrary, BMI1 knockdown in the same DTC cells suppressed their self-renewal capability. Finally, the authors demonstrated that DSF/copper-mediated BMI1 inhibition was determined by the suppression of c-Myc and E2F1 transcriptional activity on the BMI1 promoter.

The manuscript is interesting, but a number of issues should be addressed by the authors to make it suitable for publication on IJMS.

Responses: We appreciate the positive feedback from the reviewer, and the point-by-point responses are listed below.

  1. First of all, the authors limited their observations on the effect of DSF/copper on thyrosphere to only two DTC cell lines, K1 papillary thyroid carcinoma cell line and WRO follicular thyroid carcinoma cell line. They are strongly encouraged to extend their analyses to additional DTC cell lines such as TPC-1, NIM-1, FB-2, BCPAP (papillary thyroid carcinoma -PTC- cell lines) and FTC-133, FTC-236, FTC-238 (follicular thyroid carcinoma -FTC- cell lines). Extending their study on other cell lines could be useful to acquire information about the role of activated RET/PTC (TPC-1 and FB-2 cell lines) and BRAF (BCPAP cell line) on the effects mediated by DSF/copper on thyrospheres. In particular, since PTCs carrying BRAF V600E mutation are more aggressive than RET/PTC positive carcinomas, it would be interesting to analyze the ability of DSF/copper to interfere with tumor progression of more aggressive PTCs.

Responses:

We appreciate the suggestions from the reviewer. Due to the unavailability of other PTC or FTC cell lines at this time, we could not perform the suggested testing at this revision. In response to the comments, we add descriptions to present the limitations of this study in the first paragraph of the Discussion section as follows: “The present study has some limitations. It included only two DTC cell lines and lacked a xenograft mouse model of DTC to examine in vivo the anti-cancer efficacy of DSF/copper.” (page 13, line 295-297).

  1. In figure 3E the authors showed that BMI1 over-expression in K1 cells made them resistant to DSF/copper treatment. Even so data are solid, BMI1 over-expression in a cell line showing high levels of the endogenous protein (Fig. 3C) could be a bit stretched. The reviewer strongly suggests to over-express BMI1 in a BMI1-negative PTC or FTC cell line in order to compare the contribution of BMI1 to the resistance to DSF/copper respect to a BMI1-free cellular environment.

Responses:

We appreciate the suggestions from the reviewer. However, we only have K1 and WRO cells in our laboratory for this study. To respond to the comments, we add some descriptions to present the limitations of this study in the first paragraph of the Discussion section as follows: “Furthermore, the use of BMI1 non-expressing cell lines may be required to confirm the association between BMI1 and anti-cancer effect of DSF/copper.” (Page 13, line 287-298).

  1. In figure 5 c-Myc and E2F1 experiments were performed at different DSF concentrations in K1 and WRO cells: why? To make a correct and significative comparison, DSF concentrations must be the same in all experiments. In addition, in K1 cells E2F1 seems to be more sensitive than c-Myc to drug effect suggesting that the contribution of E2F1 to BMI1 transcription in K1 cells could be more relevant. To this end, the authors should perform a ChIP assay also for c-Myc in K1 cells rather than cyto/nucleus fractionation, at the same DSF concentration of Western blot depicted in panel A, and they should add 20 nM DSF in E2F1 ChIP given that Western blot showed a E2F1 decrease at 20 nM similar to 200 nM. Finally, given the possible main role of E2F1, the authors should use specific c-Myc and E2F1 inhibitors to establish the contribution of these two transcription factors to BMI1 expression in K1 cells.

Responses:

We appreciate the suggestions from the reviewer. First, although the different concentrations of DSF were used for some experiments in Figure 5, the results strongly supported the inhibitory effect of DSF/copper in c-Myc and E2F1 expression. Second, we added the ChIP data of 20 nM DSF treatment in Figure 5E, and there was a dose-dependent DSF/copper treatment's suppression of E2F1 binding on BMI1 promoter. Third, we performed the luciferase-based c-Myc reporter assay and found that DSF/copper at a concentration of 100 nM could suppress the transcriptional activity of c-Myc (Fig. 5C). Fourth, we performed the treatment of HLM006474, an E2F1 inhibitor, or 10058-F4, a c-Myc inhibitor, followed by examining the expression of BMI1 (Fig. 5E). Both inhibitors caused the downregulation of BMI1 protein expression (Fig. 5E). We also overexpressed c-Myc or E2F1 in WRO cells followed by DSF/copper treatment and found that DSF/copper lost its suppressive effect in BMI1 expression (Fig. 5G). In addition, BMI1 could be upregulated when overexpressed c-Myc (Fig. 5G). Altogether, the downregulation of c-Myc or E2F1 participates in DSF/copper-mediated BMI1 suppression in DTC cells.

  1. Which types of thyroid carcinomas were considered in figure 6? The authors do not specify if they were papillary, follicular or anaplastic carcinomas.

Responses:

According to the descriptions of the GEPIA2 website, the database of THCA is a collection of 512 papillary thyroid carcinomas (PTC) samples with 272 BRAF-like and 118 RAS-like PTCs. We have added the descriptions in the Results section (Page 10, line 249-250)

Round 2

Reviewer 1 Report

The manuscript has been significantly improved. I am happy to see it will be published in IJMS. 

Reviewer 3 Report

The authors addressed all the issues raised by the reviewer.